# Three-Dimensional Culture System: A New Frontier in Cancer Research, Drug Discovery, and Stem Cell-Based Therapy

**DOI:** 10.3390/biology14070875

**Published:** 2025-07-17

**Authors:** Guya Diletta Marconi, Antonella Mazzone, Ylenia Della Rocca, Oriana Trubiani, Jacopo Pizzicannella, Francesca Diomede

**Affiliations:** 1Department of Innovative Technologies in Medicine & Dentistry, University “G. d’Annunzio” Chieti-Pescara, via dei Vestini, 31, 66100 Chieti, Italy; antonella.mazzone@unich.it (A.M.); ylenia.dellarocca@unich.it (Y.D.R.); oriana.trubiani@unich.it (O.T.); francesca.diomede@unich.it (F.D.); 2Department of Neuroscience, Imaging and Clinical Sciences, University “G. d’Annunzio” Chieti-Pescara, Via Luigi Polacchi 11, 66100 Chieti, Italy; jacopo.pizzicannella@unich.it

**Keywords:** 3D culture system, spheroids, tumoroids, organoids, drug discovery, personalized therapy, MSCs

## Abstract

This review compares 2D with 3D model culture systems, including spheroids and organoids, highlighting the advantages and disadvantages of both systems. The goal is to provide a comprehensive overview of how 3D culture models enhance the understanding of cancer biology, drug response, and stem cell behavior. Recent findings show that 3D systems support more accurate cell/cell and cell/matrix interactions, preserve tissue architecture, and improve the predictive value of drug screening, especially for cancer and regenerative medicine applications. Spheroids and tumoroids replicate tumor heterogeneity and microenvironments, leading to a therapy that is more aligned to personalized medicine. Additionally, 3D cultures of mesenchymal stem cells, including in particular dental-derived ones, demonstrate improved differentiation and regenerative potential. In conclusion, 3D culture systems are a fundamental tool in biomedical research, bridging the gap between in vitro studies and clinical relevance; they represent a potential approach to developing treatments that are more effective while also reducing animal testing, and to advancing personalized and regenerative medicine.

## 1. Introduction

The pursuit of novel therapies has encouraged the development of novel model approaches in cancer research, drug discovery processes, and stem cell therapies. In the last decades, scientists have used the conventional two-dimensional (2D) culture system, but this type of culture does not reproduce the physiological reality. The traditional 2D culture models are not representative of real cell environments due to the loss of defined tissue organization and to cell-to-cell and cell-to-matrix interactions. Under some circumstances, the 2D system can result in cell bioactivities that do not faithfully reproduce the in vivo response [1] and dynamics of drugs [2].

Consequently, several scientists have started to use the three-dimensional (3D) culture system, which provides a highly dynamic and variable model to understand better tissue formation and function, closely reproducing the natural cell microenvironment [3,4].

The complex organization of living bodies can be reproduced by utilizing the 3D culture model, which incorporates multiple cell types and their interactions, such as cell/cell and cell/matrix connections, as well as interactions with their external environment [5].

Stem cells are unspecialized cells well known for their ability to differentiate into different cell lineages and for their self-renewal potential (Figure 1).

There are many sources of stem cells that are characterized by different differentiation potential [6]. Based on their origin, stem cells can be classified into embryonic stem cells (ESCs) and non-ESCs. Furthermore, based on their ability to differentiate into distinct cell types, these cells can be described as totipotent, pluripotent, multipotent, oligopotent, or unipotent [7].

Totipotent stem cells are typically involved in the first development stages of zygotes and are the only cells capable of dividing and differentiating into all lineages. Instead, ESCs are pluripotent stem cells derived from the inner cell mass (ICM) of embryo blastocysts; these cells are able to generate ordinary cells in germ layers, but not extra-embryonic structures [8].

Most adult stem cells are multipotent stem cells. Multipotent stem cells may differentiate into tissue derived from a single germ layer. Mesenchymal stem cells (MSCs) are multipotent stem cells, characterized by their elevated differentiative capability into osteogenic, chondrogenic, adipogenic, myogenic, and neurogenic-like lineages. Oligopotent stem cells display the restricted lineages with the differentiation capability of a specific tissue [9].

During embryogenesis, inner cell mass cells are integrated within a dynamic 3D stem cell niche, which contributes directly to both self-renewal and differentiation through the production of particular cell signaling factors, regulation of matrix stiffness, and the formation of cytokine gradients. The specific biomolecular composition of native Human Embryonic Stem Cells (hESCs)’ environment is not yet fully understood [10].

It is well known that stem cell differentiation and tissue organization are fundamental for development, organogenesis and tissue homeostasis. Organoids, self-organized three-dimensional tissue cultures that originate from stem cells, have shown a promising role in in vitro models of reproducing development, organogenesis, and tissue homeostasis events. The 3D culture methods have allowed the expansion of single stem cells into self-organizing tissues that reproduce the main features of their in vivo tissue of origin. These characteristics implicate the existence of multiple differentiated cell types and self-organization into an extremely organized tissue structure. The word organoid can be related to outgrowths from primary tissue explants (as in the mammary field) or to clonal outgrowths from single cells [11].

In cancer research, a 3D in vitro culture system has been utilized as an intermediate method between in vitro cancer cell line cultures and in vivo tumors. This 3D system has acquired importance in the cancer stem cell field using spheroids derived from the proliferation of one cancer stem cell/progenitor cell. It opens a new route for the evaluation of the pathobiology of human cancer, due to the capability of recreating in culture the 3D architecture of the tissue. For these reasons, 3D tumor cultures are considered technologies potentially useful for the development and validation of cancer treatments [12].

Furthermore, innovation in 3D technologies has gained attention in the pharmaceutical industry due to the ability of these technologies to overcome the distance existing between in vitro results and in vivo testing [13].

The great impact of the 3D culture system in the drug discovery process may be related to its exceptional features, such as high stability and long lifetime; these properties are crucial for testing the drug outcome on cellular responses [14].

Three-dimensional culture models have not only evidenced a potential role in cancer studies and drug discovery processes but also exhibited remarkable applications in the regenerative medicine field. Lastly, many scientists have focused on advancing stem cell-based and tissue regenerative tools. Ethical restrictions encourage the exploitation of adult patient-derived pluripotent stem models, which include the use of organoids due to their valuable properties.

The current work aimed to provide a comprehensive review of the advantages and limits of 3D culture models and their potential applications in cancer treatments, drug discovery, and stem cell-based therapies.

## 2. Brief History of 3D Culture Systems

The traditional 2D system has been useful in detecting cell growth and behavior, but recently, the 3D culture system has been developed to overcome the limits of the 2D monolayer system (Figure 2).

As mentioned previously, the 3D culture systems are named spheroid and organoid culture [15].

The concept of organoids dates back to the early 1900s. In 1907, it was reported for the first time by Henry Van Peters Wilson in his study on in vitro organism regeneration; he established that dissociated sponge cells may self-organize to renew an entire organism. Successively, many scientists focused their attention on the study of dissociation/reaggregation involved in the restoration of diverse types of organs, starting from dissociated amphibian pronephros and chick embryos [16]. Stem cell study acquired great importance when, for the first time, pluripotent stem cells (PSCs) were isolated and described from mouse embryos in 1981. Many years later, in 1998, the researchers were able to isolate and culture ESCs originated from blastocysts [17].

During the 20th century, due to the great impact and progress in stem cell biology, organoids became a valuable, complex, and innovative biological technology. In 2008, Yoshiki Sasai et al. described the self-organization of neural stem cells into multilayers, discovering the main regulatory factors in neural differentiation and brain formation [18]. Yoshiki Sasai was a pioneer in the organoid field with his idea of simulating 3D embryonic formation [19].

Today, scientists utilize the organoid tool to produce optic cups, pituitary gland, intestine, colon, stomach, pancreas, lung, kidney, and liver in vitro [20]. The organoid system will help scientists to understand deeply how organs develop and grow, forging a new path in human development biology.

## 3. Disadvantages, Quality Control, and Standardization in 3D Systems

The complexity of analysis, as well as the expertise, training, and technical requirements, can represent a significant challenge, since 3D structures require advanced imaging, multi-omics, or spatial transcriptomics to fully characterize cellular interactions and gradients. This represents another obstacle, since 3D structures require advanced imaging, multi-omics, or spatial transcriptomics to characterize cellular interactions and gradients fully.

Overall, 3D models are more expensive than 2D models; their assembly may require the purchasing of new laboratory products like hydrogels, scaffolds, and plastic ware as well as equipment, especially regarding imaging.

In addition, scalability is a critical issue: expanding these models for high-throughput drug screening or population-level studies often requires sophisticated bioreactors and automation technologies that are not yet widely accessible.

Furthermore, the long-term stability of 3D cultures can be difficult to achieve, as maintaining organoid viability and functionality over extended periods is technically demanding. Finally, while 3D models replicate intrahepatic architecture more realistically than 2D systems, they still have limitations in reproducing systemic effects, such as the influence of endocrine signals, the gut–liver axis, immune system dynamics, and whole-body metabolism [21].

Another important limitation consists in their low standardization and reproducibility; robust quality control and standardization measures are essential for reliable and reproducible research results of 3D cell culture systems [22].

Currently, many focuses are on harmonizing protocols for scaffold fabrication, bioprinting, cell sourcing, and analytical readouts. Moreover, best practices for batch-to-batch consistency, sterility, and validated functional assays are critical to minimize variability and enable cross-laboratory comparisons; this is essential both for selection of natural hydrogels, such as ECM, and to acquire cell culture media as well as growth factors. In fact, standardizing the protein content of the hydrogels, as well as the assay plates, is crucial to minimize differences that may arise between different batches.

In addition, standardizing the number of cells to be seeded in co-cultures and organoids is currently a challenge that can be addressed by using an automatic cell counter instead of manual counting. Furthermore, using an incubator together with gas-permeable plate seals can help prevent humidity loss and evaporation, which often occur due to the extended duration required for 3D cell culture assays.

Moreover, the use of artificial intelligence (AI) technologies has significantly improved the 3D systems’ performance for the automatization of data analysis, optimization of culture conditions, and for monitoring cellular behaviors as well as enhancing reproducibility and efficiency in biomedical research [23].

Currently, there are many different types of 3D cell-based assays; therefore, establishing clear guidelines and reference materials, including procedures, reagents, and devices as well as image and data analysis scripts will help accelerate the adoption of 3D models in preclinical workflows [24].

## 4. Spheroids in Modeling Physiological Complexity

Due to their 3D structure, spheroids display optimized biological assets in comparison with 2D single-cell arrangements.

Self-assembly reproduces the physiological events that happen during embryogenesis, morphogenesis, and organogenesis. Spheroids construct an in vivo-like microenvironment by developing more complicated cell-to-cell interactions and cell-to-ECM adhesions [25].

The biological features of spheroids include increased cell viability, stable morphology and polarization, augmented proliferative and physiological metabolic property, and stem cell differentiation.

Previous developmental studies have demonstrated that the cells in a 3D model exhibit similar cellular interactions during the developmental process and morphogenesis.

The peculiarity of 3D cell culture models is the presence of an extra spatial dimension, which allows the cells to grow in and provides a continuous engagement between the cell surface receptors and the neighboring cells. These properties will influence signal transduction and genetic expression. Due to these features, spheroids better simulate the in vivo microenvironment morphologically and physiologically than a 2D cell culture system.

Furthermore, spheroids may also be combined with other cellular aggregates to develop complex structures that can precisely simulate the features of native tissues, establishing their promising applications in the clinical field, drug testing, and disease modeling.

## 5. Spheroids in Modeling Diseases and Tumoroids

The term spheroids is commonly utilized in cancer research field, where cancer cells organize the classical multicellular tumor spheroids (MCTS) model, which has proved to be crucial for studying the biology of solid tumors.

Multicellular tumor spheroids (MCTS) typically display three concentric layers that recapitulate the cellular heterogeneity detected in solid tumors. The outer layer consists of actively proliferating cells, characterized by typical proliferation markers, including Ki-67 and Proliferating Cell Nuclear Antigen (PCNA), which are widely used to identify cycling cells. Below this zone there is the intermediate layer, where cells are mostly quiescent due to limited oxygen and nutrient supply; markers of cell cycle arrest or quiescence, such as p27^Kip1^ and p21_Cip1_, are often upregulated here. At the core, oxygen deprivation and metabolic waste accumulation create a severely hypoxic and often necrotic environment. Markers such as Hypoxia-Inducible Factor 1-alpha (HIF-1α) and Carbonic Anhydrase IX (CAIX) are commonly used to detect hypoxic cells, while cleaved caspase-3 and TUNEL staining are used to identify apoptotic or necrotic cells [26,27].

The advantages of MCTS over other 3D models are cell clonality, simplicity of maintenance, and ease of genetic manipulation. Due to these features, MCTS have been considered a promising model for high-throughput drug testing.

The cancer cells inside the spheroids interact with each other. This close cell/cell communication will influence proliferation, survival, and response to the treatment of tumor cells. Intercellular adhesion is promoted by the growth of desmosomes and dermal junctions, due to the activation of adhesion receptors such as E-cadherin, and the release of ECM proteins and proteoglycans. During spheroid development, gradients of oxygen, metabolites, nutrients, and pH are evidenced; these factors may affect the therapeutic outcomes of different drugs [28].

In the 2D monolayer system, it is not possible to mimic the diffusion-limited distribution of oxygen, nutrients, metabolites, and signaling molecules [12].

Furthermore, De Witt Hamer et al., through genomic profiling of parental tumor, monolayer, and spheroid cultures of human glioblastoma, demonstrated that spheroids are more representative of the parental tumor [29].

The 3D spheroid system has been demonstrated to be a promising tool not only for cancer research but also for studying other diseases. For example, a 3D spheroid model has been shown to closely simulate human liver function in vitro. Two-dimensional hepatocyte monolayer cultures are disposed to dedifferentiation and notably metabolic and signaling pathway alterations; indeed, the 3D culture systems maintain the in vivo features of hepatocytes keeping their phenotype and metabolic properties for longer periods. Hence, hepatic 3D spheroids are a potential platform to study better the human liver in health and disease [30].

Furthermore, it was also demonstrated that the 3D spheroid model of the blood–brain barrier (BBB) enclosing all main cell types found in the adult human brain cortex could be a helpful platform to better evaluate the function of the BBB and to comprehend the effects of chemical molecules that cross the BBB [31].

The 3D cell culture system has also been proven to be an efficient in vitro model to study neurodegenerative diseases such as Alzheimer’s and Parkinson’s disorders [32].

The main goal of cancer models is to mimic the heterogeneity intrinsic to human cancer. To address this challenge, scientists have established organotypic cancer models, organoids, also termed “tumoroids” (tumor-like organoids), that combine the 3D structureof in vivo tissues with the experimental ability of 2D cell lines (Figure 3) [33].

To date, different organoid systems have been established for a diversity of normal tissues, including the colon and small intestine, stomach, liver, mammary glands, retina, and brain [34].

In cancer research numerous patient-derived tumor organoids have been developed, including colon, prostate, gastric, breast, and pancreatic cancers, in addition to endometrial/ovary carcinomas, uterine carcinosarcoma, urothelial carcinoma, and renal carcinoma [35].

These miniaturized organoid “avatars” of a patient’s tumor simulate the histological and gene expression properties of the original tumor. The advantages of these 3D models’ are the possibility of being genetically characterized, manipulated, and expanded, making them an appealing system for leading cancer precision medicine. Organoids enable a better understanding of the tumor initiation and progression of cancers at the genetic level.

The organoids method can result in especially helpful ways to evaluate patient-specific drug sensitivities, to identify the most effective treatment strategy. Hence, this tool in the future can be used for personalized medicine, offering a suitable treatment for the singular patient.

Furthermore, relevant 3D models, such as patient-derived breast tumoroids, provided the opportunity to establish that mitochondrial transfer from stromal cells via tunneling nanotubes enhances oxidative metabolism, cancer stemness, and chemoresistance, and disrupting this exchange significantly reduces tumoroid growth [36,37].

These results showed that tumoroid systems preserve key architecture and cell–cell interactions, making them excellent models for studying mitochondrial exchange and its effects on aggressiveness, metabolism, and treatment resistance.

In the field of cancer research, 3D liver tumor models lead to the possibility of investigating “Steatotic liver disease (SLD)” with 3D liver models. Indeed, Andrea Caddeo et al. using 3D models, revealed key mechanisms driving disease progression, including lipid accumulation, chronic oxidative stress, sustained inflammation involving Kupffer cells and hepatic stellate cells, and the development of fibrosis [38].

Three-dimensional models of gastric cancer are proving valuable for research and personalized medicine. In particular, the models created with 3D bioprinting are designed to mimic the tumor microenvironment and accurately represent patient-specific characteristics. They offer a platform for studying cancer cell biology and drug resistance, and for predicting patient drug responses. Indeed, Yoo-mi Choi et al. conducted a study where they created a printed gastric cancer (pGC) model that manifested a high similarity with patients in terms of response to chemotherapy and prognostic predictability. Thus, the advent of 3D organoid culture methods offer a promising advance for preclinical studies of cancer therapy [39].

The 2D cell culture models are not capable of mimicking in vivo solid tumor events and their chemotherapy resistance [40].

The 3D models, due to their ability to reproduce the tumor microenvironment, permitted scientists to test the functionality of drugs or assessment of chemoresistance [41]. Chemoresistance in tumors is associated with the presence of inflammatory cancer-associated fibroblasts [42] and with carcinoma-associated stem cells in different types of tumors such as ovary and colorectal [43,44].

The 3D models have advanced the study of chemoresistance in cancer, leading to the development of smart nanotherapies directed to tumor stem cells in hypoxic zones, such as nanoparticles carrying retinoids and camptothecin [45] or pH/enzyme-sensitive nanocarriers that co-load chemotherapeutics (e.g., paclitaxel), anticancer stem cell agents and Programmed Death-Ligand 1 (PD-L1) inhibitors. They have also led to improved drug penetration, stimulating the immune system and reducing the number of tumor stem cells [46]. The difficulty is to establish targeted treatment strategies that are powerful in eradicating cancer stem cells, as they are resistant to anticancer drugs, leading to therapy failing, recidivism, and recurrence of tumor.

## 6. Drug Screening and Therapy Testing in Organoids/Tumoroids

Two-dimensional cell culture models are often poor predictors of in vivo drug responses, contributing to the misuse of animals’ experimentation and leading to a time-consuming and expensive drug discovery process.

Furthermore, 3D scaffolds simulate a more representative environment to evaluate cell migration, and this could be crucial to understand and prevent cancer metastasis or other disorders [47]. Moreover, this promising technology can permit the assessment of how drugs cooperate with these 3D culture models, opening a new path in the drug discovery field and innovations to personalized medicine [48].

In addition, 3D culture systems have shown high stability and extended lifetime. Previous studies established that 3D spheroids could stay in culture for up to three weeks, while 2D monolayers can stay in culture for less than a week due to the cell confluence limitations. For these reasons, tridimensional cell culture methods may be more efficient in reproducing the complexity of the tissue architecture [49]. The 3D platform, including spheroids, allows us to evaluate more accurately the mechanisms mediated by oxygen, growth factors, and nutrients that lead to the alteration of the cellular phenotype and drug effect [50].

As described earlier, tumoroids are near-physiological architectures, preserve exact properties of the original tumors, and could reliably reproduce the drug response; hence, the 3D organoid platform overcomes the distance between drug screening established with traditional 2D cell lines and clinical trials.

Several findings have already established that tumoroids can be used as a promising tool for assessing specific responses of cancer patients. Additionally, tumoroids can be useful to study specifically the epigenetic and genetic alterations triggering drug resistance [51] (Figure 4).

Based on the literature, Pauli et al. performed reasonably higher-throughput screening of 160 drugs utilizing cancer organoids from four patients, and combination drug screening for hit drugs plus 120 candidates. The data obtained by Pauli et al. established that only ~5% of the tested drugs evidenced better activity when performed in 2D versus 3D [52]. Furthermore, Jabs et al. assessed 22 drugs alone or in combination with other treatments utilizing 10 ovarian cancer organoids, establishing that the drugs tested in cancer organoids evidenced a different response in terms of genome alterations compared to drugs tested in 2D culture cells [53].

Despite being taken together, the findings of the studies mentioned above suggested the potential use of tumoroids as a tool to assess the heterogeneity in the cancer population; the clinical relevance of the data reported must be critically assessed.

Indeed, it is important to understand whether the clinical results obtained from studies on 3D systems translate into meaningful improvements in treatment efficacy or patient outcomes, which requires further in-depth investigation.

Even if the use of patient-derived tumoroids for drug testing is a promising approach, it should be noted that current clinical validation is still limited to small patient cohorts with relatively short follow-up. Larger multicenter studies and long-term clinical data are needed to fully confirm their predictive value for treatment response [54,55].

## 7. Three-Dimensional Cultures of MSCs as a Valuable Tool to Improve Therapeutic Applications in Regenerative Medicine

Three-dimensional culture systems not only have shown promising applications in drug discovery processes and in cancer studies but have also gained attention in the regenerative medicine field.

The main goal of regenerative medicine is to create an accessible and reproducible method for producing cells, tissue constructs, and/or organs.

Tissue engineering and/or regenerative medicine are areas of life science utilizing both engineering and biological fundamentals to create novel tissues and organs and to promote the regeneration of injured or diseased tissues and organs [56,57]. In tissue engineering and regenerative medicine, multicellular spheroids have been largely used as implantable therapeutics and various ex vivo tissue models [58].

Interestingly, 3D culture models represent a novel advance in simulating the in vivo microenvironment for tissue renewal. Additionally, findings on organoid cultures demonstrated that pluripotent or tissue-specific stem cells in 3D culture exhibit improved tissue-specific properties similar to in vivo tissue formation and their regenerative event [59]. Three-dimensional cultures mimic biomechanical properties of tissue, such as variable stiffness, viscoelasticity, and plasticity, which are essential to reproduce cellular behavior. Studies on viscoelasticity related to different materials such as alginates show that viscoelasticity promotes elongation, migration, proliferation, and differentiation of cells in 3D cultures [60]. Innovative studies suggest that tailoring the viscoelastic properties of biomaterials to the specific tissue you want to reproduce is critical to achieving more accurate 3D models and more translational screening results [61]. Moreover, 3D models have the advantage of integrating the mechanical plasticity of the matrix, the ability to deform, that is crucial for mimicking, for example, tumor behavior. For this reason, different materials in some studies have conferred different levels of plasticity, which has led to specific migratory capacities [62], influencing a different molecular expression of genes related to cell proliferation and mechanotransduction [63].

Wound healing is a complex, dynamic event sustained by a variety of cellular processes that need to be strictly synchronized to restore injured tissue. Stem cell spheroids reveal better therapeutic outcomes when they are transplanted into a wound healing model as a result of their enlarged growth factor release, immunomodulation properties, and target tissue integration [58,64].

MSCs, due to their features and their large distribution in several adult tissues including those of dental origin, have emerged as a promising and valuable tool for use in regenerative medicine [57,65,66]. Different studies reported that the cells derived from the oral cavity represent an incredible source of cellular therapeutics for tissue regeneration and repair [56,57,67,68]. The main feature of oral stem cells (OSCs) is their capability of self-renewal and multi-lineage differentiation. They can differentiate into a variety of cell types, including neural cells, adipocytes, chondrocytes, and osteocytes [57]. To date, six diverse human dental stem cells have been reported in the literature: human dental pulp stem cells (DPSCs), stem cells from human exfoliated deciduous teeth (SHED), human periodontal ligament stem cells (PDLSCs), human apical papilla stem cells (APSCs), human dental follicle stem cells (DFSCs), and human gingival mesenchymal stem cells (GMSCs) [69].

These cells showed the ability to adhere to a plastic surface, a fibroblast-like phenotype, and, as mentioned earlier, the ability to differentiate into osteogenic, adipogenic, and chondrogenic lineages. Moreover, they exhibited positivity for stemness surface markers and negativity for hematopoietic surface molecules [56].

In the last decades, many scientists have put their efforts into understanding the potential utilization of MSCs as a cellular treatment in different illnesses. There are many risks associated with stem cell therapies after transplantation, one of which is associated with tumorigenesis. This side effect could be related to the donor’s age, host’s tissue, growth factors expressed by the receiver tissue, and some mechanisms that regulate the MSCs’ performance in recipient tissue [70]. All these data suggest the necessity to develop and optimize the cell-based methods through the development of novel and advanced cell culture platforms.

Advantages linked to dental mesenchymal stem cells (D-MSCs) are different when compared with other sources of mesenchymal stem cells, such as bone marrow (BM-MSCs) and adipose tissue (AD-MSCs). For example, BM-MSCs require invasive and painful procedures [71], while D-MSCs can be derived non-invasively from extracted or exfoliated teeth, which are typically discarded during routine dental treatments [72]. D-MSCs have shown a higher proliferation rate as well as comparable or superior expression of pluripotency markers when compared with the other MSC lines [73]. Although AD-MSCs are relatively easy to obtain through minimally invasive liposuction, they have shown lower angiogenic and neurogenic potential compared to DPSCs [74].

Accordingly, D-MSCs remain a highly promising and ethically favorable source for tissue engineering applications [75].

Recently, an alternative and comparably easy method for enhancing the therapeutic potential of MSCs has been demonstrated, which is based on the formation and application of cells as multicellular spheroids. Three-dimensional culture systems of MSCs, such as spheroids, compared to 2D monolayer cultures of MSCs, may improve extracellular matrix (ECM) protein expression, differentiation ability, anti-inflammatory activities, and stemness [67,76].

Different studies on 3D spheroid cultures of MSCs have evidenced that the spheroid architectures can reproduce a precondition of hypoxia, which helps to study MSC treatments on ischemic tissue [77].

It was also demonstrated that 3D models of MSCs have displayed morphological alterations, increased multipotency, and exhibited increased release of several factors and anti-inflammatory molecules such as cytokines, tumor necrosis factor-α-stimulated gene/protein 6 (TSG-6), and stanniocalcin-1 (STC-1) as well as leukemia inhibitory factor (LIF), angiogenic factors, such as vascular endothelial growth factor (VEGF), and angiogenin [78].

Following the implantation of spheroids, stem cells could be stimulated to differentiate into appropriate cells for renovating the injured site. In vitro studies have shown the better differentiation potential of spheroids using the spheroid culture method in comparison with the 2D monolayer system [79].

In the last decades, scientists have focused their attention on the exploration and evaluation of the potential use of (Mesenchymal Stem Cells MSCs) as cellular therapy in several diseases. There are several side effects that can occur during stem cell treatments and a high risk of stem cell treatments correlated with tumorigenesis. Many factors can lead to the spontaneous development of tumors after MSC transplantation [80]. Three-dimensional cell aggregate cultures of MSCs evidence better properties. Recently, an alternative and easy method to enhance the therapeutic potential of MSCs has been demonstrated, which is based on the application of cells as multicellular spheroids [76]. This spheroid system may contribute to a better understanding of differentiation, tissue organization, and homeostasis of stem cells, showing a variety of approaches in tissue restoration and bone engineering [81].

As stated earlier, OSCs, due to their outstanding features, represent an incredible source in the field of regenerative medicine and tissue engineering [68]. Previous works have described the 3D spheroid formation of OSCs derived from dental pulp and root apical papilla [82]. In 2018, Moritani et al. evidenced that spheroids of human periodontal ligament stem cells (hPDLSCs), compared with 2D monolayer cultures of hPDLSC in vitro and in vivo, improved the osteogenic potential of hPDLSCs [83].

In the work published by Hyo-Jung Kim et al., they induced the formation of spheroids derived from Human Dental Follicle-Derived Stem Cells (hDFSCs), describing the potential application of a spheroid culture method for dental follicle-derived stem cells [84]. Overall, these findings underlined how a 3D culture system could be a novel and useful tool in regenerative medicine.

Regarding the regulatory context, both the Food and Drug Administration (FDA) and European Medicines Agency (EMA) increasingly recognize 3D-based drug screening models—such as organoids, spheroids, and organ-on-chip systems—as promising tools to improve the predictive power of preclinical studies and reduce animal testing, aligning with global efforts to advance New Approach Methodologies. For example, when 3D models are built using human-derived cells (e.g., Human Umbilical Vein Endothelial Cells (HUVECs), patient-derived organoids), ethical sourcing is critical. This means obtaining informed consent from donors, ensuring privacy and data protection under General Data Protection Regulation (GDPR) (EU) or Health Insurance Portability and Accountability (HIPAA) (US), and following biobank and tissue banking standards for traceability and responsible use.

Recent FDA papers, including the Three-Dimensional (3D) Cell Culture Platforms as Drug Development Tools report (2021) [85] and the 2024 concept paper on integrating NAMs [86], emphasize the need for robust validation, reproducibility, and clear context of use, supported by initiatives like the Drug Development Tool (DDT) Qualification Program and collaborative workshops with industry consortia (e.g., the IQ MPS workshop). Similarly, the EMA’s EU-IN Horizon Scanning Report on NAMs (2025) [87] and its Regulatory Science to 2025 strategy [88] highlight the importance of horizon scanning, standardization, and early scientific advice to facilitate the adoption of innovative 3D models in regulatory submissions. Both agencies stress that developers must demonstrate that 3D models reliably replicate human physiology for defined purposes, comply with Good Laboratory Practices where relevant, and engage early with regulators to align models with qualification frameworks such as the FDA’s Drug Development Tool (DDT) and EMA’s Qualification of Novel Methodologies (QoNM) programs, ensuring these complex in vitro systems can be confidently used to support safety and efficacy assessments in drug development.

## 8. Future of 3D Models

The future of 3D models is growing with their integration with advanced technologies such as bioprinting, organ-on-chip devices, and artificial intelligence (AI)-assisted analysis, which together promise to enhance the physiological relevance, scalability, and predictive power of in vitro models.

Bioprinting enables the precise fabrication of complex, multicellular tissue structures with defined architecture and vascularization, which bridges the gap between simple spheroids and fully functional tissue mimetics. Organ-on-chip systems further advance this by recreating dynamic, microfluidic environments that simulate human organ functions and interactions in real time, providing more accurate data on drug absorption, distribution, metabolism, and toxicity [89].

Meanwhile, AI and machine learning tools allow researchers to analyze vast and complex datasets generated by these advanced models, uncovering subtle biological patterns, improving model reproducibility, and accelerating hypothesis generation. AI has profoundly influenced the field of 3D cell cultures by facilitating automated data processing, forecasting models, and live tracking of cellular activities. Machine learning techniques assist in refining culture parameters, improving consistency and productivity in biomedical studies [22].

Together, these technologies position 3D models at the forefront of next-generation drug discovery, personalized medicine, and regulatory science, offering ethically responsible, human-relevant alternatives that align with the growing global push to replace and reduce animal testing.

## 9. Conclusions

In conclusion, 3D culture systems provide an exceptional in vitro model, enabling the experimentation of cellular responses that faithfully reproduce the in vivo environment. These advanced tools could be extremely useful in cancer research, drug discovery, and stem cell-based therapy.

## Figures and Tables

**Figure 1 biology-14-00875-f001:**
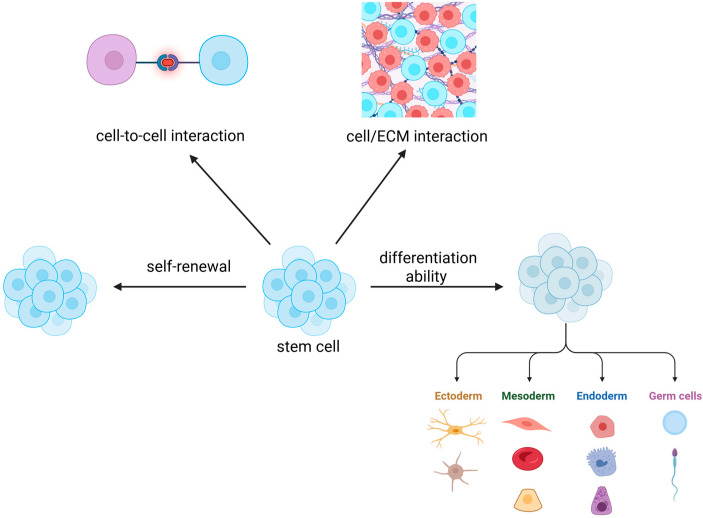
Fundamental features of stem cells: mechanisms of self-renewal, intercellular communication, extracellular matrix (ECM) interactions, and differentiation potential (Created with Biorender.com).

**Figure 2 biology-14-00875-f002:**
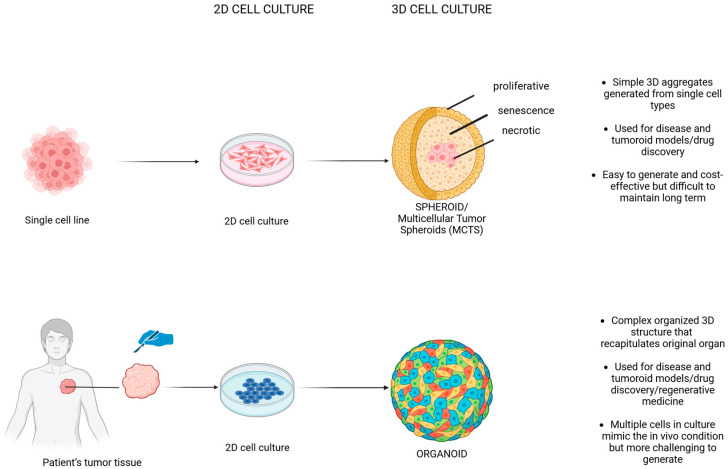
Comparison of 2D and 3D cell culture systems: advantages and limitations (Created with Biorender.com).

**Figure 3 biology-14-00875-f003:**
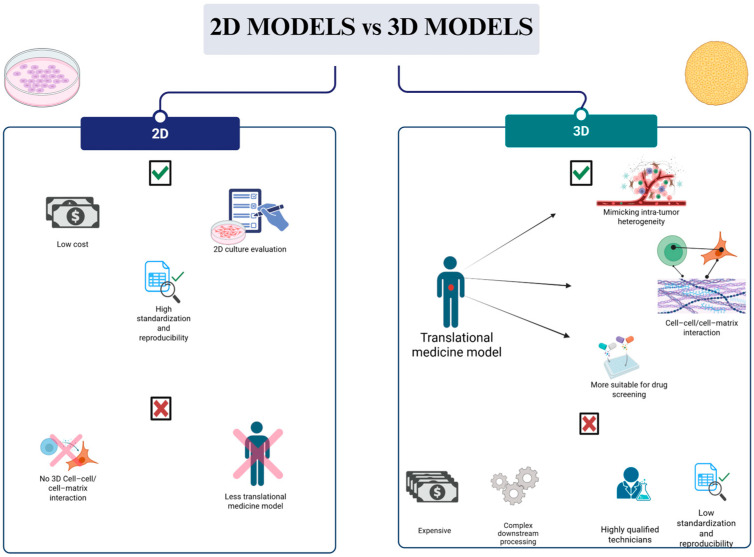
Three-dimensional biological models: spheroids versus organoids—origins, structural complexity, and biomedical applications (Created with Biorender.com).

**Figure 4 biology-14-00875-f004:**
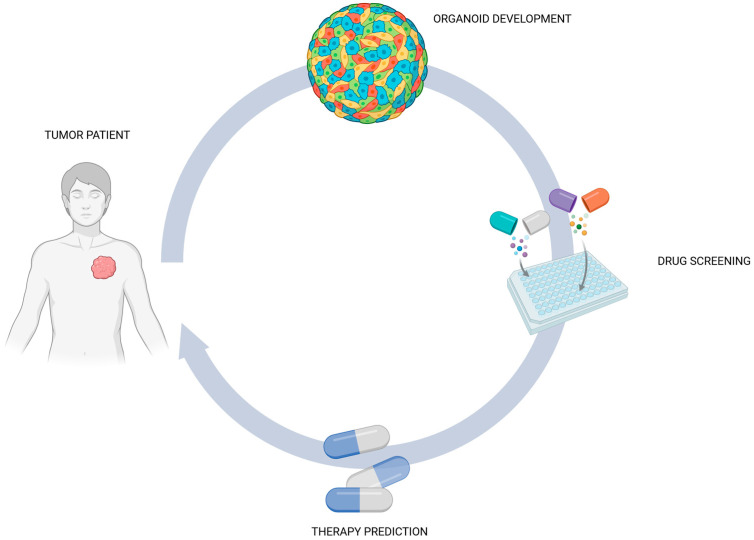
Organoids as a platform for drug screening: from miniaturized organ modeling to predictive personalized therapy (Created with Biorender.com).

## Data Availability

Data are contained within the article.

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
