# Peer review of "Three-Dimensional Culture System: A New Frontier in Cancer Research, Drug Discovery, and Stem Cell-Based Therapy"

_biology, 2025, doi:10.3390/biology14070875_

Round 1
Reviewer 1 Report
Comments and Suggestions for Authors
Dear editor,
The manuscript entitled “3D culture systems: a new frontier in cancer research, drug discovery and stem cell-based therapy’’ provides a general overview of current 3D models used in biomedical research and outlines their advantages over traditional 2D monolayer culture systems.
While the paper highlights many relevant benefits of 3D systems, particularly in cancer research, some sections lack depth, and several ideas are repeated across different parts of the manuscript which affects the clarity and structure of the content. To help improve the overall quality and scientific value of the manuscript, I suggest the following points:
- Throughout the manuscript the authors use several terms interchangeably: tumoroids, organoids, avatars, mini organs, to refer to 3D models. It is important to stick to accurate and consistent terminology, especially in the 3D culture field where nomenclature is widely agreed upon. For 3D models of solid tumors, the correct term is multicellular tumor spheroids (MCTS), not MTS as mentioned in the manuscript.
‘’Organoids’’ is a broader term that generally refers to miniaturized and simplified versions of organs, not tumors. Based on recent consensus at international meetings (e.g. Paris 3D culture conference), it's encouraged to refer to models using their specific names such as ‘’3D liver model’’ or ‘’3D skin model’’. Please also avoid using the term ‘’mini organ’’, which is not scientifically appropriate. - The manuscript lacks sufficient referencing in several areas. Please add proper citations for the following statements:
- Line 57: ‘’and their internal (e.g. cell-cell, cell-matrix) or external (e.g, cell-environment) interactions’’
- Line 62-63: ‘’There are many sources of stem cells characterized by different differentiation potential’’
- Line 66: ‘’these cells can be described as totipotent, pluripotent, multipotent, oligopotent or unipotent’’
- Line 183-184: ‘’The exclusive cytoarchitecture of cells inside the MTS system reproduces cell proliferation, morphology, oxygenation, nutrient and drug absorption’’
- Line 318-319: ''
- In Section 3 (lines 155-157), the explanation regarding U-bottom plates needs to be reconsidered. While it's true that cells aggregate in ultra-low attachment plates, they do not replicate their complex 3D architecture seen in scaffold-based systems. This part may be misleading and should be removed or reworded, especially since the next paragraph already explains the spatial dimensions of 3D models more accurately.
- Section 4 is split into two different headings without a clear structure. Consider combining both parts under a single title or using subsections (e.g. 4.1, 4.2) for clarity
- In line 230, the authors mentioned that 3D cultures recapitulate the histological and gene expression profiles of the tumor of origin. This is a key point and would benefit from further development. The three classical layers of MCTS should be described: what are the proliferation markers in the outer layer, the markers of cell quiescence in the intermediate zone, and hypoxia/necrosis markers in the core?
- In line 241, the mention of chemoresistance needs clarification. In fact, hypoxic zones within tumors, both in vivo and in vitro, often contain cancer stem cells, which are known to resist standard therapies and contribute to relapse. Expanding the section would strengthen the argument. You could also mention emerging therapeutic approaches being tested on 3D models such as nanotherapies and immunotherapeutic combinations.
- In line 279 the authors refer to a ‘’different response’’ between 2D and 3D models. It would be helpful to elaborate: are we talking about changes in gene expression, phenotypic behavior or transcriptomic profiles?
- Similarly, in line 297, the authors refer to ‘’improved tissue specific properties in 3D models’’. This should be expanded to include how 3D cultures better replicate features of the native tissue microenvironment such as mechanical stiffness, viscoelasticity, and other biomechanical properties relevant to cellular behavior.
With these revisions, the manuscript will better reflect the current state of the field and provide readers with a clearer, more scientifically grounded understanding of the role and value of 3D models in modern biomedical research.
Provided the authors adequately address the minor revisions suggested in this report, I recommend this manuscript for publication in your journal.
Comments on the Quality of English Language
English grammar needs improving and in vivo / in vitro should be written in italic
Reviewer 2 Report
Comments and Suggestions for Authors
Dear Dr. Marconi and colleagues,
I have carefully read your manuscript on 3D culture systems and believe it adds real value to the discipline, to the point of considering it worthy of publication. You have managed to integrate diverse applications in areas such as oncology research, drug development, and regenerative medicine, creating a coherent and useful account for both beginners and experienced experts.
Positive and Strong Points for the Publication There are several aspects in which your review particularly stands out. The way you present the historical evolution, from Wilson's pioneering work in 1907 to the most current applications, provides a context that, frankly, is often lacking in other recent reviews. Furthermore, the clarity with which you differentiate between spheroids, organoids, and tumoroids helps dispel many common misunderstandings in the literature, making it truly educational.
I was intrigued by the focus on tooth-derived stem cells, which provides an original perspective and sets your review apart from similar ones. The balance you strike between fundamental studies and recent advances is very appropriate, and the way you structure the text, from basic principles to clinical applications, makes complex concepts easy to understand.
Aspects to Review or Further Introduce
Although the advantages of 3D systems are fairly comprehensive, I think you could strengthen your critical analysis of current limitations. There are significant challenges in terms of standardization and reproducibility, which perhaps merit further review. For example, protocols for generating spheroids often result in heterogeneous structures in size and architecture, and this ultimately complicates comparisons between different studies.
Your reflection on the use of tumoroids as patient "avatars" is very interesting, but clinical validation is still limited. The studies you mention (Pauli, Jabs, etc.) are encouraging, although studies with more patients and long-term follow-up are still needed to confirm their true usefulness. On the other hand, although you mention the difference in pharmacological activity between 2D and 3D (around 5%), the clinical significance of this data deserves a more detailed analysis.
Regarding the economics and scalability, I think there is an important gap in your review. 3D technologies typically require a significant investment, both in equipment and staff training, and this can limit their implementation in laboratories with fewer resources.
Minor comments and suggestions
It might be helpful if you included a brief section on quality control and standardization initiatives in 3D systems. It would also be very helpful to address the regulatory context for 3D-based drug screening models, as this would give even more clinical weight to the work.
The section on dental mesenchymal stem cells is interesting, but you could enrich it by comparing them in more detail with other sources of MSCs, addressing aspects such as practical advantages, availability, and ethical issues.
Finally, the future of the field lies in integration with emerging technologies such as bioprinting, organ-on-chip devices, and artificial intelligence-assisted analysis. I think this horizon is worth exploring further.
Final Evaluation
Overall, your work perfectly fulfills the objective of offering a comprehensive and up-to-date overview of 3D culture systems. You have managed to show how these technologies can revolutionize biomedical research and their potential for different applications. The content is well-founded, with appropriate references and a clear structure.
In my opinion, the manuscript deserves acceptance, with minor points to investigate deeper into issues of critical analysis and standardization. These changes do not require extensive revision but rather add some nuances. I sincerely believe that your review will serve as a useful reference for many researchers who want to make the leap from 2D to 3D and will be an important contribution to the advancement of personalized and regenerative medicine.
Regards,
Reviewer 3 Report
Comments and Suggestions for Authors
This mini Review is of general interest. Its content is almost up to dated and clear.
The manuscript addresses a highly relevant and timely topic in biomedical research: the advantages of 3D cell culture systems over traditional 2D models in cancer research, drug discovery, and stem cell-based therapies, provides a comprehensive overview, starting with a brief history of 3D culture systems and then delving into specific applications like spheroids in modeling physiological complexity and diseases (including tumoroids). It also highlights their role in drug screening and regenerative medicine, with a particular focus on mesenchymal stem cells (MSCs) and dental-derived MSCs. Moreover it outlines practical advantages, such as improved predictive value for drug screening, potential for personalized medicine, reduction of animal testing, and enhanced differentiation and regenerative potential of stem cells.
The paper is logically structured with clear sections covering the introduction, history, specific applications (spheroids, tumoroids, MSCs), and conclusions.
Pitfalls:
Some information, particularly the comparison between 2D and 3D systems and the general advantages of 3D cultures, is repeated across the Simple Summary, Abstract, Introduction, and subsequent sections. For example, the idea that 3D systems overcome limitations of 2D models by mimicking the in vivo environment is stated multiple times. This could be condensed for better flow and conciseness.
There are several instances of awkward phrasing or minor grammatical errors that could be improved with careful proofreading (e.g., "Truthfully, their systems support..." , "The highly organization of body could be reproduced..." , "does not outcome in dedifferentiation" , "may has potential applications" , "non-suitable drugs to access the in vivo assays" ).
While the text mentions figures (e.g., Figure 1: "Stem cells and their main characteristics", Figure 2: "Advantages and disadvantages of 2D and 3D culture systems"), I do not have access to the actual image content. Based on their titles, they appear to be generic illustrations. For a review, more specific or novel diagrams illustrating complex mechanisms or experimental setups (if applicable and cited) would enhance the manuscript's value.
While the manuscript focuses on the advantages of 3D models and the disadvantages of 2D models, a more balanced review would include a dedicated section or more in-depth discussion of the current limitations and challenges associated with 3D culture systems themselves (e.g., standardization, cost, scalability, complexity of analysis, long-term stability in all contexts, limitations in replicating systemic effects). Although it mentions "disadvantages of both the systems" in the Simple Summary, the detailed text predominantly focuses on 2D limitations.
The concerns are related to the too long historical and 2D models that must be consistently reduced and the excessively short para #4 called. Tumoroids: Models of Cancer
In this short paragraph the Authors would describe the use and application of Organoids to cancers.
Albeit it is a field in development they should have described in more details at least some Results obtained and the perspectives in some types of cancers like Breast /with mitochondrial transfer and related aggressiveness; Lung cancer, gastric cancer or others, including the main Results and related paper citations.
This paragraph is of great impact and must, therefore, be enlarged.
Overall, the manuscript presents a valuable and informative review on the growing field of 3D cell culture systems. Addressing the repetitive content, refining the language, and including a more critical discussion of 3D model limitations would significantly strengthen the manuscript.
Comments on the Quality of English Language
English must be consistently amended.
Round 2
Reviewer 3 Report
Comments and Suggestions for Authors
The Authors correctly answered to all my previous comments.